

# Effects of using mobile augmented reality for simple interest computation in a financial mathematics course

Laura Alicia Hernández Moreno[1], Juan Gabriel López Solórzano[1], María Teresa Tovar Morales[1], Osslan Osiris Vergara Villegas[2] and Vianey Guadalupe Cruz Sánchez[2]

[1] Facultad de Contaduría Pública y Administración, Universidad Autónoma de Nuevo León, Monterrey, Nuevo León, México
[2] Instituto de Ingeniería y Tecnología, Universidad Autónoma de Ciudad Juárez, Juárez, Chihuahua, México

## ABSTRACT

Understanding the concept of simple interest is essential in financial mathematics because it establishes the basis to comprehend complex conceptualizations. Nevertheless, students often have problems learning about simple interest. This paper aims to introduce a prototype called "simple interest computation with mobile augmented reality" (SICMAR) and evaluate its effects on students in a financial mathematics course. The research design comprises four stages: (i) planning; (ii) hypotheses development; (iii) software development; and (iv) design of data collection instruments. The planning stage explains the problems that students confront to learn about simple interest. In the second stage, we present the twelve hypotheses tested in the study. The stage of software development discusses the logic implemented for SICMAR functionality. In the last stage, we design two surveys and two practice tests to assess students. The pre-test survey uses the attention, relevance, confidence, and satisfaction (ARCS) model to assess students' motivation in a traditional learning setting. The post-test survey assesses motivation, technology usage with the technology acceptance model (TAM), and prototype quality when students use SICMAR. Also, students solve practice exercises to assess their achievement. One hundred three undergraduates participated in both sessions of the study. The findings revealed the direct positive impact of SICMAR on students' achievement and motivation. Moreover, students expressed their interest in using the prototype because of its quality. In summary, students consider SICMAR as a valuable complementary tool to learn simple interest topics.

# INTRODUCTION

The economic factor is involved in practically all the processes of making decisions. Therefore, to avoid making wrong financial decisions, it is recommendable to know how money is obtained, managed, invested, and optimized. The lack of these skills could be solved by completing a financial education course (*Carpena & Zia, 2020*).

Corresponding author
Osslan Osiris Vergara Villegas,
overgara@uacj.mx

Financial education must start at an early stage. *Berry, Karlan & Pradhan (2018)* and *Sun et al. (2020)* demonstrated how financial education helped prevent problems such as having low credit scores or defaulting on a loan. Because of the relevance of financial education, the United States of America included various finance courses as a part of the primary school curriculum (*Urban et al., 2020*). Other countries such as China (*Ding, Lu & Ye, 2020*), Ghana (*Berry, Karlan & Pradhan, 2018*), Hong Kong (*Feng, 2020*), and India (*Carpena & Zia, 2020*) successfully adopted this trend; however, success cannot be generalized. As pointed out by *Arceo & Villagómez (2017)* and *Bruhn, Lara & McKenzie (2014)*, underdeveloped countries such as México reported minimum benefits due to the inclusion of financial education in schools.

To obtain insights about why students show no interest in financial education, we performed monitoring of undergraduates enrolled in financial mathematics courses at four public northern Mexican universities. All the students, no matter what program they are enrolled in, must take a financial mathematics course because it is mandatory within the school curriculum. Therefore, we monitored students from the accounting, administration, business, and engineering fields. As a result, we detected three problems: (i) students lack mathematical skills; (ii) sometimes the techniques used by the professors to teach the basics are boring, and (iii) students do not comprehend the basics such as simple and compound interest, which are fundamental to sound financial education.

In financial mathematics, interest is the cost of using the money of a person or an institution. If somebody borrows money, then interest must be paid. Contrary, if somebody lends money, then interest is earned. Interest is calculated as simple or compound; the former is a percentage of the principal amount of a loan, whereas the latter accrues and is added to the accumulated interest of previous periods; therefore, it includes interest on interest (*Hastings, 2015*).

*Abylkassymova et al. (2020)* and *Blue & Grootenboer (2019)* focused their research on seeking alternative methods to solve students' difficulties in understanding the basic concepts explained in a financial mathematics course. The most used options are individualized explanations outside of class time, multimedia material, computer simulations, and information and communications technologies (ICTs). Nevertheless, there are still opportunities to propose teaching-learning strategies to help students comprehend financial education basics. This paper assesses mobile augmented reality (MAR) technology as an alternative learning strategy to comprehend simple interest topics. *Gutíerrez et al. (2016)* and *Chen (2019)* defined MAR as "a real-time direct or indirect view of a real-world environment that has been augmented by adding virtual computer-generated information to it". In summary, mobile augmented reality is a novel way of superimposing digital content into the real context.

*Akçayır & Akçayır (2017)* and *Arici et al. (2019)* explained the benefits of mobile augmented reality in educational settings, especially for mathematics. The benefits include student achievement increase, autonomy facilitation (self-learning), generation of positive attitudes to the educational activity, commitment, motivation, knowledge retention, interaction, collaboration, and availability for all.

Motivated by MAR advantages and the problems detected regarding financial education, this paper aims to develop the simple interest computation with mobile augmented reality (SICMAR) prototype and to assess its effects in an undergraduate financial mathematics course. The study, divided into pre and post-test, was designed to assess students' motivation, achievement, technology acceptance, and prototype quality.

The main contributions of the paper are summarized below.

1. We explain the details to develop the SICMAR prototype.
2. We offer a proposal to assess students' motivation, achievement, technology acceptance, and SICMAR quality in a real educational setting.
3. We explain the facts to support that SICMAR could be a valuable complementary tool to learn about simple interest.

The rest of the paper is organized as follows. 'Learning Mathematics with Augmented Reality' discusses related work about AR to support mathematical learning. In 'Research Design', the basis to develop SICMAR and the surveys created are described. The results obtained from tests and the corresponding discussion are shown in 'Results'. Finally, conclusions are outlined in 'Discussion'.

## LEARNING MATHEMATICS WITH AUGMENTED REALITY

Many research studies have been published on AR usage for educational purposes. Interested readers can consult the works by *Akçayır & Akçayır (2017)*; *Saltan & Arslan (2017)*; *Garzón, Pavón & Baldiris (2019)* and *Arici et al. (2019)* to obtain a comprehensive overview of the educative fields that have been addressed.

As *Arici et al. (2019)* explained, most AR works focused on sciences such as medicine, physics, history, arts, and astronomy; however, the social fields, laws, and business are less addressed. *Ibáñez & Delgado-Klos (2018)* presented a literature review of AR to support science, technology, engineering, and mathematics (STEM) learning. *Medina, Castro & Juárez (2019)* and *Demitriadou, Stavroulia & Lanitis (2020)* presented comparisons between virtual reality and augmented reality for mathematics learning. In both studies, no significant difference was found between virtual and augmented reality technologies in contributing to mathematics learning. The work of *Radianti et al. (2020)* is recommended as a starting point if the readers want to explore the field of virtual reality in education.

For this study, we found papers related to augmented reality for mathematics teaching-learning. However, only studies published between 2013 and 2020 were considered. The query strings included "mathematics", "financial mathematics", "augmented reality", "mobile augmented reality", "teaching", "education", and "learning". Also, we use the Boolean operators "OR", "AND" to mix multiple strings. We collected the papers from journals included in the journal citation reports (JCR) and manuscripts published in conferences through the Web of Science (WoS).

As a result, we detected 17 studies focused on learning mathematics inside formal and informal environments. Concerning the formal settings, the learners' education level ranges from preschool to undergraduate. The elementary level is the one in which more studies have been published. Moreover, geometry is the subject with more implementations. This

is due to the ability of augmented reality to promote interaction and visualization with 2D and 3D objects.

*Salinas et al. (2013)* tested the impact of AR on learning algebraic functions using 3D visualizations. The experience was assessed by 30 undergraduates from Mathematics I course. Likewise, *Barraza, Cruz & Vergara (2015)* used AR to help undergraduate students learn quadratic equations. The pilot study was conducted with 59 students at a Mexican school, and most comments obtained were positive. An AR app for mathematical analysis was presented by *Coimbra, Cardoso & Mateus (2015)*. Thirteen undergraduates participated in the experience, where most of them expressed "*classes should all be like this*". Regarding geometry, *Gutíerrez et al. (2016)* presented an AR system aimed at the learning of descriptive geometry. A positive impact on the spatial ability of 50 undergraduates was found.

*Purnama, Andrew & Galinium (2014)* designed an AR tool to help elementary students learn the protractor's use. According to the students' responses, 92% found that the prototype makes the learning process faster than using a conventional method. *Li et al. (2017)* designed an augmented reality game for helping elementary students in the counting process. The two students who participated in the experience expressed that learn to count was easy using AR. Moreover, *Tobar, Fabregat & Baldiris (2015)* and *Cascales et al. (2017)* explained the advantages of using mobile augmented reality to learn mathematics in elementary special education needs (SEN) contexts.

*Sommerauer & Muller (2014)* conducted a pre-test and post-test with 101 participants at a mathematics exhibition. The aim was to measure the effect of AR on acquiring and retaining mathematical knowledge in an informal learning environment. The pre-test score captured previous knowledge regarding the mathematical exhibits, while the post-test captured the knowledge level after visiting the exhibition. The results revealed that visitors performed significantly better on post-test questions.

A summary of the features of the papers analyzed is shown in Table 1. There are no signs of papers related to financial mathematics, neither for simple interest computation. Regarding the preferred software for implementing AR, Vuforia is the leader. The number of participants varies from 2 to 140. It seems that there is no consensus about the sample size to validate an AR study. Only five works presented assessments about students' motivation. Most of the work was concentrated on prototype perception and students' achievement. No work focused on assessing technology acceptance was found. According to the theory base employed, qualitative research was the most used (seven times), followed by the nonparametric Wilcoxon signed-rank test (four times). Most of the technologies used to implement the prototypes were mobile devices, which evidences PCs are less preferred, and smart glasses are not yet used in academic scenarios, mainly due to the high cost. All the works introduced single-user-based applications because it is still complex to build collaborative applications. Based on the analysis conducted, our proposal's novelty relies on the field addressed (simple interest) and the constructs assessed in the same study (motivation, achievement, quality, and technology acceptance).

**Table 1  Summary of 17 experimental augmented reality studies focused on learning mathematics.**

| Author(s) | Software | APP Name | Sample | Subject to learn | Learners | Assessment | Theory base |
|---|---|---|---|---|---|---|---|
| *Salinas et al. (2013)* | N/A | TEAM | 30 | Algebraic functions | Undergraduate | Prototype perception | Qualitative research |
| *Sommerauer & Muller (2014)* | Aurasma studio | Mathematics exhibition | 101 | Mathematics | Mathematics exhibition visitors | Knowledge retention | Wilcoxon test |
| *Purnama, Andrew & Galinium (2014)* | Open CV | ARGLT | N/A | Geometry | Elementary | Achievement | Percentages |
| *Estapa & Nadolny (2015)* | Layar creator | Mathematics instruction | 61 | Dimensional analysis | High school | Achievement and motivation | F-test, and ARCS |
| *Barraza, Cruz & Vergara (2015)* | Vuforia SDK | pARabola | 59 | Quadratic equations | Undergraduate | Prototype perception | Qualitative research |
| *Tobar, Fabregat & Baldiris (2015)* | Nyartoolkit | Gremlings in my mirror | 20 | Mathematical logic | Elementary | Achievement | Qualitative research |
| *Coimbra, Cardoso & Mateus (2015)* | N/A | AR an enhancer for math | 13 | Mathematical analysis | Undergraduate | Learning increase | Qualitative research |
| *Gutíerrez et al. (2016)* | Vuforia SDK | DiedricAR | 50 | Descriptive geometry | Undergraduate | Spatial ability improvement | Percentages |
| *Cascales et al. (2017)* | N/A | Augmented book | 22 | Money managing | Elementary | Achievement and motivation | Wilcoxon test |
| *Rohendi, Septian & Sutarno (2017)* | Artoolkit | AR geometry media | N/A | Geometry | High school | Prototype perception | Qualitative research |
| *Li et al. (2017)* | Vuforia SDK | See me roar | 2 | Counting | Elementary | Prototype perception | Qualitative research |
| *Aulia & Muhimmah (2018)* | Vuforia SDK | DorDor | 140 | Counting | Elementary | Prototype perception | Qualitative research |
| *Cai et al. (2019)* | N/A | Seven, Super spaces, Magic coins | 101 | Probability and statistics | High school | Conceptions, approaches, and self-efficacy | ANCOVA |
| *Gecu & Delialioglu (2019)* | N/A | Augment | 72 | Geometric shapes | Preschool | Understanding | Mann–Whitney U and Wilcoxon test |
| *Chen (2019)* | Hp Reveal | MobileAR | 82 | Algebra and Geometry | Elementary | Motivation and math anxiety | ANCOVA and ARCS |
| *Ibáñez et al. (2020)* | N/A | ARGEO | 93 | Geometry | High school | Achievement and motivation | ANOVA and ARCS |
| *Sarkar, Kadam & Pillai (2020)* | ARCore | ScholAR | 27 | Geometry | Middle school | Perspectives, approaches, and motivation | ANOVA and ARCS |
| Our proposal (2020) | Vuforia SDK | SICMAR | 103 | Simple interest | Undergraduate | Motivation, quality achievement, technology acceptance | ARCS, Wilcoxon test, $t$-test, and TAM |

## RESEARCH DESIGN

In this research, we use a mixed-method to allow the synergistic usage of qualitative and quantitative data (*Reeping et al., 2019*). Furthermore, this research is considered exploratory and descriptive. Exploratory because we investigate the problem in an early stage and obtain insights into what is happening. Descriptive, because we describe the features of the phenomenon studied. The systematic methodology to conduct the research comprises four stages: (i) planning; (ii) hypotheses development; (iii) software development; and (iv) design of data collection instruments.

### Planning

After the conversation with three financial mathematics professors, the planning stage began. The professors agreed with us about the three problems detected during the monitoring. However, they mentioned the barriers that students face when learning simple interest: (a) the problem is not analyzed, therefore, it is not understood; (b) they confuse simple interest with compound interest and vice versa; (c) the terms involved to solve the computation are wrongly cleared; (d) the concepts such as principal, amount, interest rate, and time are misinterpreted; and (e) conversions between time units are wrongly performed. Professors explained that around 70% of Mexican students commit at least one of the errors mentioned above. They also stated that simple interest knowledge is fundamental for mastering finances and understanding complex concepts such as compound interest, amortization tables, and annuities. Hence, students must comprehend the topic.

We ask professors for an explanation to understand how to compute simple interest. The explanation was based on the following example. When an individual borrows money, the lender expects to be paid back the loan amount plus an additional charge for using the money called interest. In contrast, when money is deposited in a bank, it pays the depositor to use the capital, also called interest.

*Simple interest (I)* represents the fee you pay on a loan or income you earn on deposits. In other words, simple interest represents the price of the money over a specific period. As shown in Eqs. (1) and (2), there are two ways to compute simple interest. Furthermore, notice the four terms involved: (i) *Principal (P)* is the original sum of money borrowed (also called present value); (ii) *Interest rate (r)* is a fraction/percentage of the principal (charged) per unit of time; (iii) *Time (t)* represents the time period over which the interest rates apply/are charged; and (iv) *Amount (A)* is the total accrued amount (principal plus interest), represents the future value of the financial operation (*Hastings, 2015*).

$$I = Prt, \tag{1}$$

$$I = A - P. \tag{2}$$

The terms regarding principal, interest rate, time, and the amount can be cleared from Eqs. (1) and (2) as is depicted in Eqs. (3), (4), (5), and (6), respectively.

$$P = \frac{I}{rt}. \tag{3}$$

$$r = \frac{I}{Pt}. \tag{4}$$

$$t = \frac{I}{Pr}. \tag{5}$$

$$A = P + I. \tag{6}$$

In Eqs. (1) to (6), it is common to use years as the time unit. However, time could also be expressed in days, weeks, fortnights, months, bimesters, quarters, or semesters. For any calculation, if the period for $r$ and $t$ is defined in different units, then a conversion must be computed, which often causes mistakes.

To this end, we propose a prototype to support simple interest learning and design two surveys and two practice tests to assess the effects of using it with undergraduate students. The prototype is called "simple interest computation with mobile augmented reality (SICMAR)".

### Hypotheses development

Our study assesses students' motivation when learning simple interest in traditional settings and with SICMAR. Also, we assess the students' achievement when learning in both settings through a test. Finally, we consider obtaining insights about SICMAR technology acceptance and quality. Thereby, we pose twelve hypotheses.

## Students motivation

Motivation affects what, how, and when the learners learn, and it is directly related to the development of students' attitudes and persistent efforts toward achieving a goal (*Lin et al., 2021*). Motivation is an activity that must be performed to (i) attract and sustain students' attention (A); (ii) define the relevance (R) of a content students need to learn; (iii) help students to believe they succeed in making efforts (gain confidence (C)); and (iv) assist students in obtaining a sense of satisfaction (S) about their accomplishments in learning (*Cabero-Almenara & Roig-Vila, 2019*). In this sense, Keller's ARCS model provides guidelines for designing and developing strategies to motivate students learning (*Li & Keller, 2018*).

In previous studies, the ARCS model was used to observe if mobile augmented reality could be a resource that motivates students to learn anatomy and art (*Cabero-Almenara & Roig-Vila, 2019*), dimensional analysis (*Estapa & Nadolny, 2015*), and geometry (*Ibáñez et al., 2020*), obtaining promising results. Therefore, the present paper poses the following five hypotheses.

**H₁**: there is a significant difference in students' attention scores in the pre-test and the post-test.

**H₂**: there is a significant difference in students' relevance scores in the pre-test and the post-test.

**H₃**: there is a significant difference in students' confidence scores in the pre-test and the post-test.

**H₄**: there is a significant difference in students' satisfaction scores in the pre-test and the post-test.

**H₅**: there is a significant difference in students' motivation scores in the pre-test and the post-test.

### Students achievement

Academic achievement is the extent to which a student has accomplished specific goals that focus on activities in instructional environments (*Bernacki, Greene & Crompton, 2019*). In this paper, student achievement is related to how capable the students are when solving simple interest computation problems. The studies by *Purnama, Andrew & Galinium (2014)*; *Estapa & Nadolny (2015)*; *Tobar, Fabregat & Baldiris (2015)*; *Cascales et al. (2017)*, and *Ibáñez et al. (2020)* show how the use of MAR can affect mathematics academic achievement positively. Therefore, we propose the following hypothesis.

**H₆**: students learning with SICMAR achieve higher scores in simple interest tests than students exposed to traditional learning.

### Technology acceptance

The technology acceptance model (TAM) was formulated by *Davis (1989)*. TAM suggested that the perceived ease of use (PEU) and the perceived usefulness (PU) are determinants to explain what causes the intention of a person to use (ITU) a technology. The perceived ease of use refers to the degree to which a person believes that using a system would be free from effort. The perceived usefulness refers to the degree to which the user believes that a system would improve his/her work performance. The intention to use is employed to measure the degree of technology acceptance (*Davis, 1989*).

In previous research, TAM was used to examine the adoption of augmented reality technology in teaching using videos (*Cabero-Almenara, Fernández & Barroso-Ozuna, 2019*) and learning the Mayo language (*Miranda et al., 2016*). However, no studies related to the use of TAM in mathematical settings were detected. In this paper, the TAM is extended with prototype quality variable to explain and predict the SICMAR usage. The aim is to study the relationships between quality, perceived ease of use, and perceived usefulness and their positive effects on students' intention to use SICMAR.

The family of statistical multivariant models that estimate the effect and the relationships between multiple variables is known as structural equation modeling (SEM) (*Al-Gahtani, 2016*). Therefore, we use SEM to test the following hypotheses.

**H₇**: quality positively affects students' perceived usefulness of SICMAR.

**H₈**: quality positively affects students' perceived ease of use of SICMAR.

**H₉**: perceived ease of use positively affects students' perceived usefulness of SICMAR.

**H₁₀**: perceived ease of use positively affects students' intention to use SICMAR.

**H₁₁**: perceived usefulness positively affects students' intention to use SICMAR.

The words ''positively affect'' mean that when the measured value of one variable increases, the related variable also increases.

## SICMAR quality

Software quality is the field of study that describes the desirable characteristics of a software product. Establishing a measure for the quality of the software is not an easy task. However, attributes such as design, usability, operability, security, compatibility, maintainability, and functionality can be considered to define metrics (*Dalla et al., 2020*). When the quality of an AR product is evaluated, questions such as how fast the system responds, how difficult it is to manipulate the system and markers, and to which extent the illumination affects marker recognition must be answered (*De Paiva & Farinazzo, 2014*). In this paper, the quality was assessed considering the design and usability of SICMAR as recommended by *Barraza, Cruz & Vergara (2015)* and *Pranoto et al. (2017)*.

In the literature, no MAR works that report a hypothesized minimal mean value of quality that serves for comparison were found. Therefore, we propose the following procedure: (i) determine the minimum and the maximum length of the Likert scale; (ii) compute the scale range by subtracting ($5-1 = 4$) and dividing by five ($4/5 = 0.80$); (iii) add the range to the least scale value to obtain the maximum. The ranges computed for a five-point Likert scale are $1-$—$1.8$–$2.6$-—$3.4$–$4.2$–$5$. Results greater than 3.4 and less or equal to 4.2 are considered as good quality. Thus, the mean value of 3.8 was supposed to determine good quality. Moreover, as is explained in the results section, this value is the median obtained after experimentation. Hence, the following hypothesis is established.

**H$_{12}$**: the mean value evaluated by students regarding the quality of SICMAR is greater than 3.8.

### Software development

We consider the cascade model to establish the stages to develop SICMAR. The cascade model is a linear procedure characterized by dividing the software development process into successive phases (*Ruparelia, 2010*). The model encompasses five phases, including (i) requirements; (ii) design; (iii) implementation; (iv) testing; and (v) maintenance. A visual representation of the SICMAR phases is shown in Fig. 1.

## Requirements

According to *Billinghurst, Grasset & Looser (2005)*, the physical components of the interface (inputs), the virtual visual and auditory display (outputs), and the interaction metaphors must be considered to build intuitive AR applications. Therefore, we determine five characteristics of the prototype to deal with the barriers faced by students: (i) a set of markers will be used to determine the term to compute and the parameters involved (inputs); (ii) 2D models will represent all the information needed for the calculations; (iii) markers' movement will be used to observe the 2D models from different perspectives; (iv) the calculation to solve will be defined with a combination of markers (touch manipulation metaphor); and (v) we will employ virtual objects, such as text boxes, arrows, and images, to explain step-by-step calculations (outputs). A traditional computer application cannot offer all these features.

From the variety of information and communication technologies, PCs and mobile devices were considered to implement the prototype due to the high probability that a

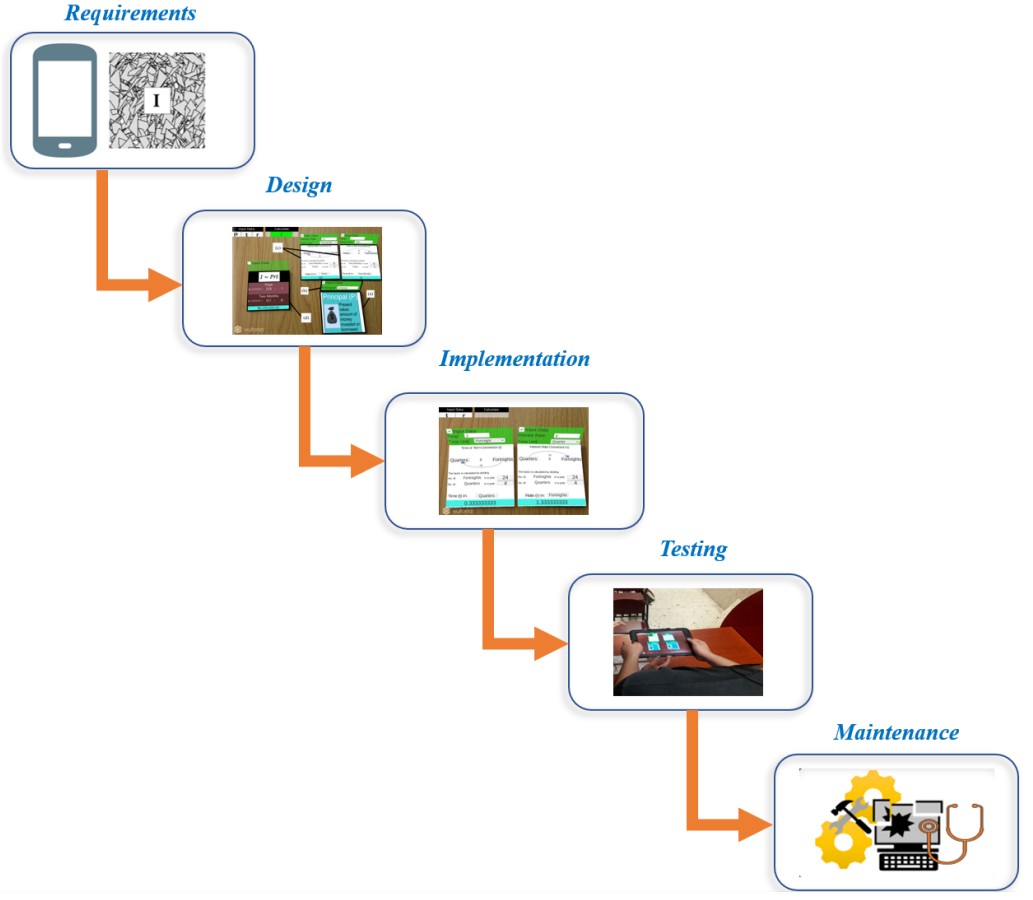

**Figure 1    Visual representation of the methodology to develop SICMAR.**

student has either one of them. The main differences between PCs and mobile devices are the display size, how it is manipulated, the processing power, bandwidth, and usage time. Portability, sensors included, and ease manipulation were considered to select mobile devices. Indeed, young users prefer mobile devices because they can be used anytime, carried from place to place, and connected to the Internet all day long. Moreover, recent studies have shown that almost 75% of AR works for educational settings were implemented on mobile devices obtaining satisfactory results (*Bernacki, Greene & Crompton, 2019*; *Cabero-Almenara, Fernández & Barroso-Ozuna, 2019*). AR applications are different from conventional applications that use a mouse and keyboard. By superimposing virtual information, mobile augmented reality increases perception (virtual objects can be observed from different perspectives) and user interaction (objects are manipulated with the fingers) with the environment; a non-AR application cannot offer that feature. Also, MAR can turn a classic learning process into an engaging experience because students perceive learning as a game (*Arici et al., 2019*).

Regarding mobile devices, Android and iOS-based devices are the leaders. Nevertheless, we selected Android because (i) it is the leading mobile operating system worldwide; (ii)

the price for publishing an app in the play store is much lesser than posting an app in the apple store; (iii) the cost of Android-based devices is less than iOS-based devices; due to price, a student rarely has an iPhone; (iv) it possesses a good support architecture and functional performance; (v) the customization level offered makes it easy to use; (vi) the assortment of the batteries' sizes overpowers the iPhone (_Ivanov, Reznik & Succi, 2018_).

## Design

There are various alternatives to implement AR solutions, including Wikitude, ARToolKit, Augumenta, Easy AR, HP Reveal, and Vuforia, offering exciting characteristics. Considering the analysis presented in Table 1 and based on the authors' experience, Vuforia Software Developer Kit (SDK) was selected. Vuforia is a robust platform that contains the libraries to implement the tasks related to AR, including real-time marker detection, recognition and tracking, and the computations for object superimposition. Unity 3D was employed to create the SICMAR visual environment and all the 2D virtual objects that will be superimposed on each marker.

SICMAR was designed based on the framework proposed by _Barraza, Cruz & Vergara (2015)_. The framework comprises four subsystems (i) the rendering; (ii) the context and world model; (iii) the tracking; and (iv) the interaction, which works together to create the mobile application.

In the rendering subsystem, two main tasks are executed: (i) displaying the video acquired from the real world, and (ii) rendering the 2D models. We designed a touch-based graphical user interface (GUI) to display the components and the video acquired from the mobile device. At the top of the GUI, we inserted two sections: (i) input data and (ii) calculate (output). The first show the input terms (markers) detected inside the scene, and the second shows the term the user wants to compute (see the upper left corner in Fig. 2). We used the Unity sprite renderer for rendering all the photorealistic images of the 2D models that will be superimposed inside the real-world video stream.

The context and world model subsystem includes the design of image targets (markers), the data about the interest points, and the 2D objects that are going to be used in the augmentation. We used the Brosvision marker generator to design the markers to represent each of the five terms explained in (1) to (6). As shown in Fig. 3, the markers include lines, triangles, quadrilaterals, and at the center, a square with a letter corresponding to the simple interest term was added. Using Vuforia, we conducted a test of the contrast-based features (interest points) of the individual markers visible to the camera. All the markers earned five stars rating, which means they included excellent features for detection and tracking. Finally, four 2D objects were created for user interactions and augmentations, as shown in Fig. 2. (a) To display information about the detected marker (input term). (b) To capture the user inputs and determine if a term is handled as input or output. (c) To display information about the time conversions. (d) To display the calculation result (output term) or show an error.

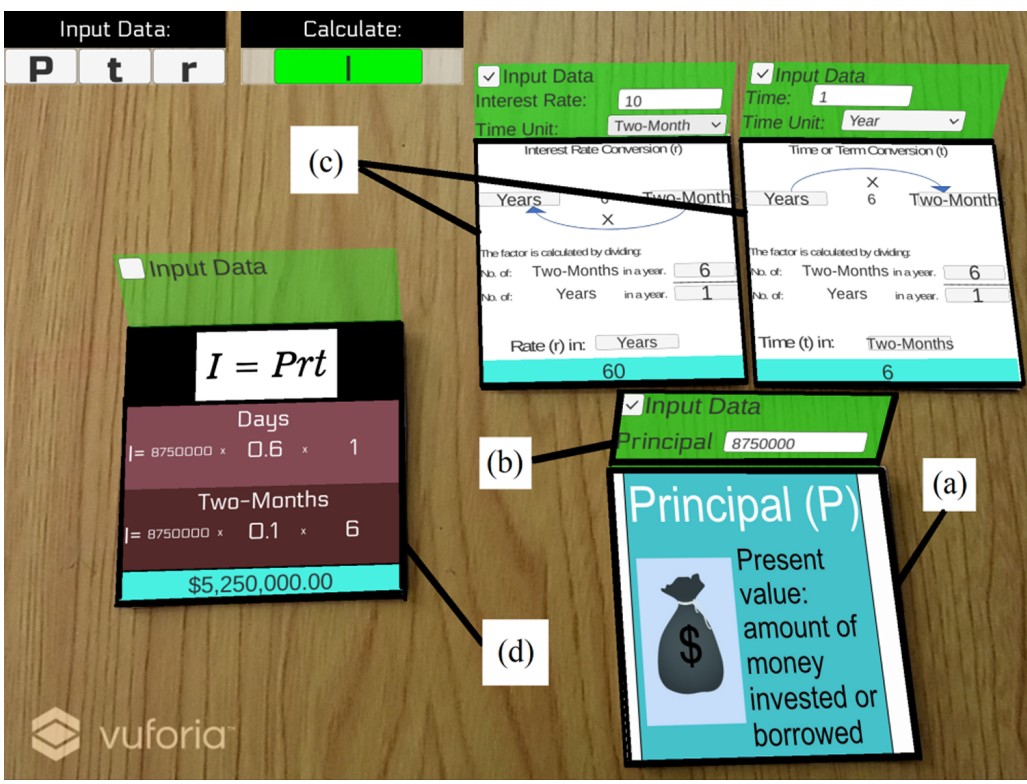

**Figure 2** The screen for simple interest computation: (A) Objects to display information, (B) Interaction controls, (C) Objects to show conversions, and (D) Objects to show a result.

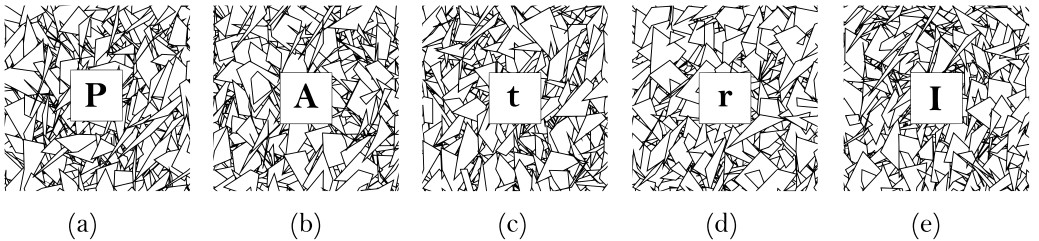

**Figure 3** The set of five SICMAR markers: (A) Principal, (B) Amount, (C) Time, (D) Interest rate, and (E) Simple interest.

Vuforia SDK was used to carry out the tracking subsystem. This subsystem exchanges marker tracking information with the rendering subsystem to superimpose the virtual 2D objects to the original scene displayed to the user.

Finally, the interaction subsystem collects and processes any input required by the user. A series of C# scripts were linked to the GUI objects. When a tap occurs on the screen, verification is carried out to determine if an element was touched. If the verification is

valid, the search for a marker starts. If a marker is detected, then the corresponding method is invoked to carry out the task.

## Implementation

The logic implemented to solve any of the Eqs. (1) to (6) is the following. The user taps the SICMAR icon to start the execution. The presentation screen is displayed, and the camera of the mobile device is turned on. When the user shows a valid marker in the front of the camera, it is recognized as the desired output. Then, the position, rotation, and perspective of the marker are computed, and the corresponding virtual object is superimposed accordingly to the view of the real scene. Next, the input checkbox is activated, and the prototype waits for the user to show the markers for input terms. When input markers are recognized, the text boxes to insert data are displayed, and the 2D objects are superimposed inside the real scene. Any marker different from the first selected can be used as input. The user must insert the data for each term with the keyboard of the device. Once the data was introduced, the input checkbox must be disabled to perform the computation. Immediately, verification is conducted to detect if the necessary data for the computation were inserted correctly. If there is any missing data, an error object is displayed, else the output calculated is presented. The process can be executed continuously.

## Testing

An example of simple interest computation using SICMAR is shown in Fig. 2. If the user inserts $r$ and $t$ with different periods, then the associated conversions are computed. Notice that the result of simple interest computation is highlighted with the color blue. As shown in Fig. 4, the user selected quarters for $r$ and fortnights for $t$. At the bottom of the screen, the value obtained from the conversion is displayed and explained for both periods. An example of students testing SICMAR is shown in Fig. 5.

## Maintenance

Two experienced software developers tested our beta version of SICMAR. They recommended conducting modifications related to the color and size of the objects. We also performed modifications to the GUI, including object location changes and those related to interactivity. As explained in the discussion section, the students recommended performing additional modifications to SICMAR, which will be implemented soon.

### *Design of data collection instruments*

We designed two surveys to collect the data. The first serves to obtain information about students' motivation when the professor explained the simple interest topic using traditional materials (textbooks, slides, and whiteboards). The second survey gathers data about students' motivation when learning with SICMAR, technology acceptance, and prototype quality. Besides, we designed a data consent form and two five-item tests to measure students' achievement. It is essential to highlight that Spanish was the language employed for the surveys and the whole experiment. Therefore, in this paper, English translations are presented.

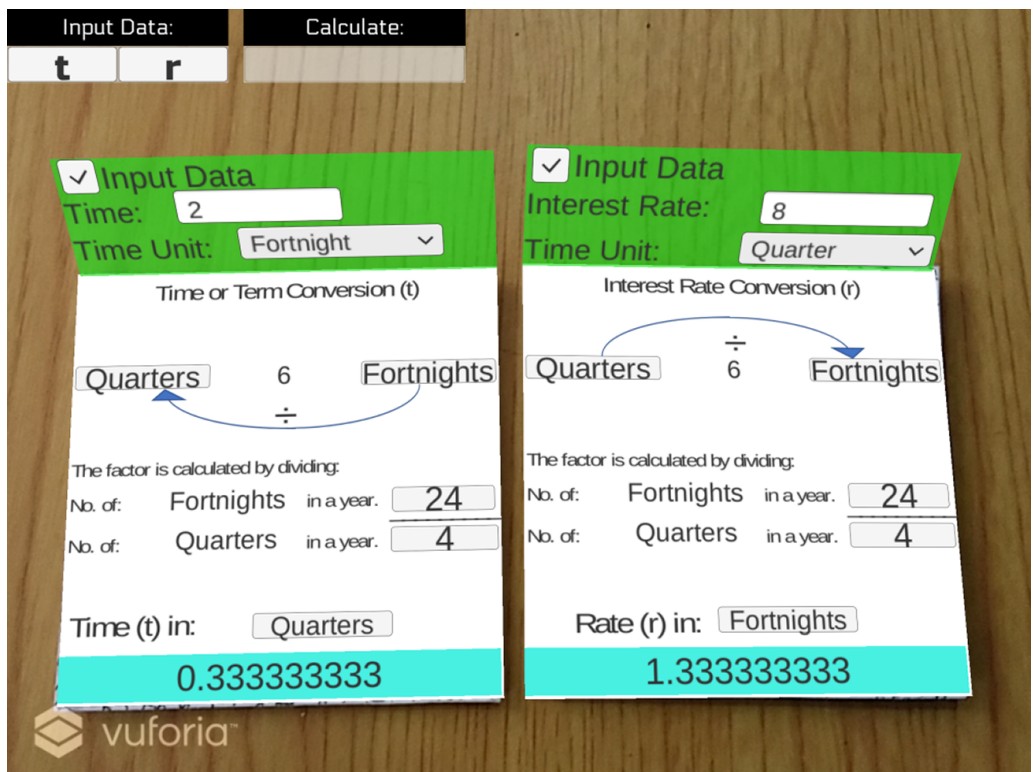

**Figure 4** An example of the conversion of *r* and *t* terms.

## The first survey (pre-test)

The first survey has two sections: the first includes items to collect students' general information, such as name, gender, and age, and the second includes items related to Keller's ARCS motivation model (*Li & Keller, 2018*).

The instructional materials motivation survey (IMMS) assesses students' motivation based on the ARCS model that includes 36 items distributed as 12 items for (A), nine items for (R), nine items for (C), and six items for (S). Although IMMS was used and tested with a Cronbach $\alpha = 0.96$, it is long, and not all items are necessary, especially those measured in a negative or reverse way (*Chen, 2019*). Therefore, the reduced IMMS (RIMMS) proposed by *Loorbach et al. (2015)* was employed. RIMMS comprises 12 five-point Likert scale items, three for each ARCS dimension. The original version was translated and adapted to the lesson of simple interest (see the left side of Table 2). The minimum score on the RIMMS survey is 12, and the maximum is 60 with a midpoint of 36.

The items about attention measure the degree to which the professor's lesson attracts the learner's attention. We consider the organization, quality, and variety of the materials employed. On the other hand, we use the content and style of explanations to measure the lesson's relevance perceived by students. The items regarding confidence measure the degree to which the learner felt confident while completing the simple interest lesson. The

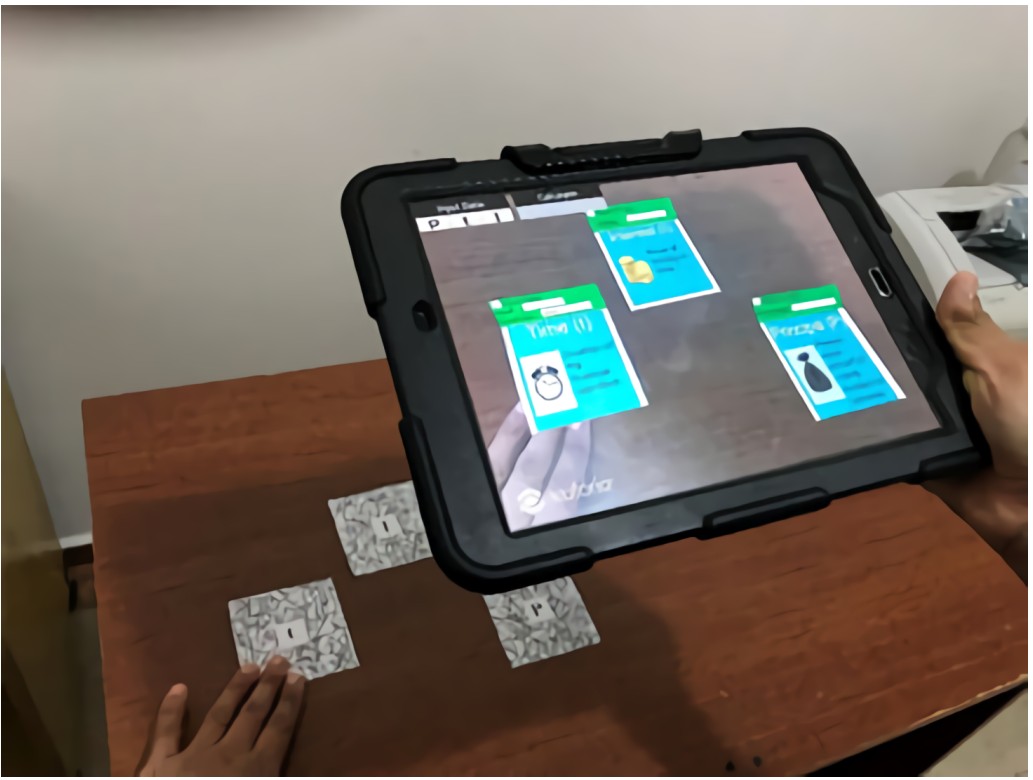

**Figure 5** **Students testing SICMAR prototype.**

final three items measure the degree to which the learner finds the lesson satisfactory and the intention to keep working.

## The second survey (post-test)

The second survey comprises four sections. The first includes items to collect students' general information. The second section includes the 12 RIMMS items of the first survey but adapted to assess students' motivation using SICMAR (see the right side of Table 2).

The third section related to TAM comprises four items for perceived usefulness, five for perceived ease of use, and two for the intention to use (see Table 3). The 11 items used a five-point Likert scale and were adapted from *Miranda et al. (2016)*. The minimum score for TAM is 11, and the maximum is 55 with a midpoint of 33. The four items regarding perceived usefulness measure the extent to which students believe that SICMAR would improve their performance in learning simple interest. The easiness that students perceived when using SICMAR is measured with the five items related to perceived ease of use (prototype manipulation employing the markers). The last two items measure the degree of acceptance when students use SICMAR.

The fourth section aims to gather information about SICMAR quality. The ten items based on the Likert five-point scale were adapted from *Barraza, Cruz & Vergara (2015)*. We collect information about SICMAR design (colors, size of the objects, velocity) and

**Table 2  The first survey (pre-test) and the first part of the second survey (post-test).**

| General Data | | | | | |
|---|---|---|---|---|---|
| **Name (s):** | | | **Surname:** | | |
| **Age:** | | | | | |
| **Gender:** | o (Male) | | o (Female) | | |

| **ARCS Professor** | | | **ARCS SICMAR** | | |
|---|---|---|---|---|---|
| Please think about each statement concerning the professor's lesson you have just participated and indicated how true it is. Give the answer that truly applies to you, not what you would like to be true or what you think others want to hear. Use the following values to indicate your response to each item: 1 = N*ot true*, 2 = *Slightly true*, 3 = *Moderately true*, 4 = *Mostly true*, and 5 = *Very true*. | | | Please think about each statement concerning the SICMAR you have just used and indicated how true it is. Give the answer that truly applies to you, not what you would like to be true or what you think others want to hear. Use the following values to indicate your response to each item: 1 = *Not true*, 2 = *Slightly true*, 3 = *Moderately true*, 4 = *Mostly true*, and 5 = *Very true*. | | |
| | **Mean** | **SD** | | **Mean** | **SD** |
| **Attention (A)** | **3.95** | **0.81** | **Attention (A)** | **4.14** | **0.81** |
| A1. The quality of the materials used helped to hold my attention. | 3.91 | 0.80 | A1. The quality of the contents displayed helped to hold my attention. | 4.19 | 0.93 |
| A2. The way the information was organized helped keep my attention. | 3.97 | 0.89 | A2. The way the information was organized (buttons, menus) helped keep my attention. | 4.09 | 0.90 |
| A3. The variety of readings, exercises, and illustrations helped keep my attention on the explanations. | 3.98 | 1.04 | A3. The variety of 2D models and interactions helped keep my attention on the explanations. | 4.17 | 0.94 |
| **Relevance (R)** | **3.87** | **0.74** | **Relevance (R)** | **4.38** | **0.70** |
| R1. It is clear to me how the content of this lesson is related to things I already know. | 3.35 | 1.02 | R1. It is clear to me how the content of SICMAR is related to things I already know. | 4.48 | 0.81 |
| R2. The content and style of explanations convey the impression that being able to work with simple interest is worth it. | 4.05 | 0.92 | R2. The content and style of explanations used by SICMAR convey the impression that being able to work with simple interest is worth it. | 4.31 | 0.86 |
| R3. The content of this lesson will be useful to me. | 4.22 | 0.89 | R3. The content of SICMAR will be useful to me. | 4.36 | 0.86 |
| **Confidence (C)** | **3.87** | **0.77** | **Confidence (C)** | **4.08** | **0.73** |
| C1. As I worked with this lesson, I was confident that I could learn how to compute simple interest well. | 4.12 | 0.91 | C1. As I worked with SICMAR, I was confident that I could learn how to compute simple interest well. | 4.07 | 0.88 |
| C2. After working with this lesson for a while, I was confident that I would be able to pass a test about simple interest. | 3.54 | 0.94 | C2. After working with SICMAR for a while, I was confident that I would be able to pass a test about simple interest. | 4.08 | 0.92 |
| C3. The good organization of the content helped me be confident that I would learn about simple interest. | 3.96 | 0.83 | C3. The good organization of SICMAR helped me be confident that I would learn about simple interest. | 4.11 | 0.75 |
| **Satisfaction (S)** | **3.80** | **0.77** | **Satisfaction (S)** | **4.10** | **0.83** |
| S1. I enjoyed working with this lesson so much that I was stimulated to keep on working. | 3.61 | 0.89 | S1. I enjoyed working with SICMAR so much that I was stimulated to keep on working. | 3.92 | 0.93 |
| S2. I really enjoyed working with this simple interest lesson. | 3.85 | 0.87 | S2. I really enjoyed working with SICMAR. | 4.07 | 0.92 |
| S3. It was a pleasure to work with such a well-designed lesson. | 3.95 | 0.82 | S3. It was a pleasure to work with such a well-designed prototype. | 4.31 | 0.89 |
| **ARCS** | **3.87** | **0.69** | **ARCS** | **4.17** | **0.66** |

usability (results obtained) that together determined quality, as shown at the bottom of Table 3.

**Table 3** **The third and fourth sections of the second survey (post-test).**

<table>
<tr><td colspan="5" align="center">SICMAR TAM</td></tr>
<tr><td colspan="5">Please select the number that best represents how do you feel about SICMAR acceptance: 1 = Strongly disagree, 2 = Disagree, 3 = Neutral, 4 = Agree, 5 = Strongly agree.</td></tr>
<tr><td></td><td>Mean</td><td>SD</td><td>Standardized factor loadings</td><td>Hypotheses interpretation</td></tr>
<tr><td>Perceived Usefulness (PU)</td><td>4.09</td><td>0.80</td><td></td><td></td></tr>
<tr><td>PU1. I could improve my learning performance by using SICMAR</td><td>3.97</td><td>0.86</td><td>0.762</td><td><0.01, Accepted</td></tr>
<tr><td>PU2. I could enhance my simple interest proficiency by using SICMAR</td><td>3.99</td><td>0.97</td><td>0.771</td><td><0.01, Accepted</td></tr>
<tr><td>PU3. I think SICMAR is useful for learning purposes.</td><td>4.25</td><td>0.93</td><td>0.820</td><td><0.01, Accepted</td></tr>
<tr><td>PU4. Using SICMAR will be easy to remember the concepts related to the calculation of simple interest.</td><td>4.17</td><td>0.97</td><td>0.832</td><td><0.01, Accepted</td></tr>
<tr><td>Perceived Ease of Use (PEU)</td><td>4.04</td><td>0.81</td><td></td><td></td></tr>
<tr><td>PEU1. I think SICMAR is attractive and easy to use</td><td>3.79</td><td>1.13</td><td>0.679</td><td><0.01, Accepted</td></tr>
<tr><td>PEU2. Learning to use SICMAR was not a problem for me due to my familiarity with the technology used.</td><td>4.32</td><td>0.97</td><td>0.805</td><td><0.01, Accepted</td></tr>
<tr><td>PEU3. The marker detection was fast.</td><td>4.02</td><td>1.04</td><td>0.664</td><td><0.01, Accepted</td></tr>
<tr><td>PEU4. The tasks related to the manipulation of controls were simple to execute.</td><td>3.92</td><td>1.04</td><td>0.817</td><td><0.01, Accepted</td></tr>
<tr><td>PEU5. I was able to locate the areas for conversions and calculations quickly.</td><td>4.19</td><td>0.86</td><td>0.792</td><td><0.01, Accepted</td></tr>
<tr><td>Intention to Use SICMAR (ITU)</td><td>4.38</td><td>0.82</td><td></td><td></td></tr>
<tr><td>ITU1. I want to use the app in the future if I have the opportunity.</td><td>4.28</td><td>0.96</td><td>0.925</td><td><0.01, Accepted</td></tr>
<tr><td>ITU2. The main concepts of SICMAR can be used to learn other topics.</td><td>4.49</td><td>0.81</td><td>0.754</td><td><0.01, Accepted</td></tr>
<tr><td>TAM</td><td>4.12</td><td>0.72</td><td></td><td></td></tr>
<tr><td colspan="5">SICMAR Quality</td></tr>
<tr><td colspan="5">Please select the number that best represents how do you feel about SICMAR quality: 1 = Not at all, 2 = A little, 3 = Moderate ly, 4 = Much, 5 = Very much.</td></tr>
<tr><td>Quality questions</td><td>Mean</td><td>SD</td><td>Standardized factor loadings</td><td>Hypotheses interpretation</td></tr>
<tr><td>Q1. SICMAR showed all the concepts explained by the teacher.</td><td>4.45</td><td>0.84</td><td>0.450</td><td><0.01, Accepted</td></tr>
<tr><td>Q2. The results obtained with SICMAR were correct.</td><td>4.24</td><td>0.82</td><td>0.562</td><td><0.01, Accepted</td></tr>
<tr><td>Q3. The colors used for conversions were adequate.</td><td>4.17</td><td>0.91</td><td>0.527</td><td><0.01, Accepted</td></tr>
<tr><td>Q4. The texts and numbers displayed by SICMAR were legible.</td><td>4.13</td><td>0.94</td><td>0.627</td><td><0.01, Accepted</td></tr>
<tr><td>Q5. The size of the buttons allowed the easy manipulation of SICMAR.</td><td>3.16</td><td>1.22</td><td>0.531</td><td><0.01, Accepted</td></tr>
<tr><td>Q6. SICMAR velocity of response to carry out the calculations was fast.</td><td>4.40</td><td>0.85</td><td>0.528</td><td><0.01, Accepted</td></tr>
<tr><td>Q7. The classroom illumination was adequate.</td><td>3.79</td><td>0.98</td><td>0.513</td><td><0.01, Accepted</td></tr>
<tr><td>Q8. The manipulation of the electronic device I use was straightforward.</td><td>3.76</td><td>1.00</td><td>0.676</td><td><0.01, Accepted</td></tr>
<tr><td>Q9. Markers' manipulation was easy.</td><td>3.65</td><td>1.05</td><td>0.747</td><td><0.01, Accepted</td></tr>
<tr><td>Q10. The manipulation of the device in conjunction with the markers was easy.</td><td>3.56</td><td>1.06</td><td>0.703</td><td><0.01, Accepted</td></tr>
<tr><td>Quality</td><td>3.93</td><td>0.62</td><td></td><td></td></tr>
</table>

## The practical tests

Two financial mathematics professors helped us to design a set of practical exercises regarding simple interest computation. We divided the set of exercises to design two practical tests with five items each. The first test is applied after professor intervention, and the second after using SICMAR. Professors carefully reviewed both tests to ensure similar difficulty. In both tests, the first two items ask the students to compute simple interest. The

last three questions are challenging because the terms to compute must be cleared from Eqs. (1) and (2). The third question deals with principal computation, while the fourth and fifth deal with interest rate and time calculation, respectively. An example of two pre-test exercises is shown on the left column of Table S1, while the post-test examples are shown on the right column.

## RESULTS

The study was conducted in early March 2020. A classroom at a public university located in northern México was used as an educational setting. One professor that participated in the planning stage organized the sessions that comprised the study. Both sessions were conducted with three days of difference.

The Mexican university where the study was conducted imposed three restrictions regarding the participation of the students and professor: (i) all students enrolled in the financial mathematics course must participate; (ii) the professor could only use the time established in the curriculum to offer explanations, and (iii) only one session could be used to test SICMAR. Therefore, we decide to conduct a quasi-experimental study to establish a cause-and-effect relationship between independent and dependent variables.

A quasi-experimental study is characterized because the sample to study is not selected randomly, and control groups are not required. Instead, participants are assigned to the sample based on non-random criteria previously established (all the students in the financial mathematics course must participate). This study is also called a nonrandomized or pre-post intervention and is frequently employed to conduct research in the educative field (*Otte et al., 2019*).

Before the experiment, students did not have prior knowledge of the concepts related to simple interest. Students were informed about the research goal and that the data obtained will be treated with confidentiality and used only for academic purposes. Moreover, students completed a consent form regarding data use. Institute of Engineering and Technology of Universidad Autonoma de Ciudad Juárez emitted the approval to use the data and reviewed the consent form students filled out.

In the first session, which lasted two hours, the professor explained the simple interest lesson employing traditional materials. Students were then asked to realize a practice consisting of the pre-tests five test exercises and fill out the first survey. At the end of the first session, we request students to get an Android-based mobile device for the second session.

The second session lasted one and a half hours and started with an explanation about the use of SICMAR. Afterward, each student received a set of markers. Fortunately, all students brought the Android mobile device. Hence, mobile devices (smartphones and tablets) with different features were used, which allow us to observe the variety of devices in which SICMAR can be executed. The average time to interact with the prototype was 39 min. Next, students were asked to realize the post-test practice consisting of five exercises and fill out the second survey. Students answered the surveys through the Internet (Microsoft forms) and practical exercises on a sheet of paper in both sessions.

**Table 4  Cronbach's alpha values for both surveys.**

| Measurement | $\alpha$ |
| --- | --- |
| A | 0.867 |
| R | 0.679 |
| C | 0.821 |
| S | 0.872 |
| **ARCS (pre-test)** | **0.934** |
| A | 0.847 |
| R | 0.776 |
| C | 0.814 |
| S | 0.889 |
| **ARCS (Post-test)** | **0.931** |
| PU | 0.877 |
| PEU | 0.859 |
| ITU | 0.815 |
| **TAM** | **0.921** |
| **Quality** | **0.839** |

### Preliminary data analysis

Due to the restrictions imposed by the university, students were not divided into a control and experimental group. One hundred thirty-nine students enrolled in the financial mathematics course were surveyed. Data collected from the surveys was downloaded from Microsoft forms to create a database with IBM SPSS software. The responses obtained were minutely revised. The extreme values were not discarded, but registers with incomplete information were identified. The registers with incomplete information correspond to 36 students who did not attend the second session. Therefore, the final sample comprises data from 103 students.

The sample size was deemed valid due to (i) our sample almost doubled the mean ($M = 58.2$) from the fourth column of Table 1; and (ii) the section related to ARCS is the biggest of our surveys; therefore, the rule of thumb that a sample should have at least five times as many observations as there are variables to be analyzed was fulfilled (5x12 = 60).

Of the 103 participants, $n = 59$ (57.28%) were female, and $n = 44$ (42.72%) were male. The participants' ages ranged from 18 to 30 years with a mean age ($M = 19.74$, $SD = 1.93$). We measure the internal reliability of the surveys with Cronbach's Alpha ($\alpha$). A summary of the results is shown in Table 4. Values greater than 0.7 are accepted (good-excellent). The total item correlation computed does not reflect the necessity of eliminating any item. Therefore, the $\alpha$ value for R in pre-test ARCS was also accepted.

### Assessment of students motivation with RIMMS

This part of the study allowed us to assess if a significant difference in motivation is obtained when comparing the professor's lesson and SICMAR. The mean and standard deviation for each item are displayed in Table 2. All scores exceed the central value of the scale. Moreover, the greater mean values were always obtained with SICMAR. The minimum difference is observed for attention ($4.14 - 3.95 = 0.19$) and the maximum for relevance

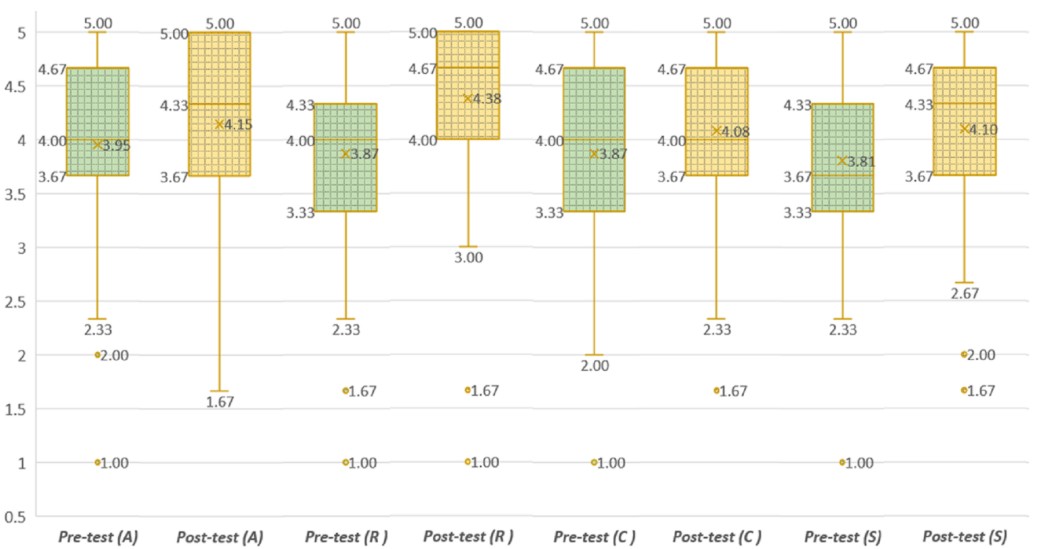

**Figure 6** Results for ARCS pre-test and post-test.

(4.38−3.87 = 0.51). The difference for the whole study is (4.17−3.87 = 0.3). The results for both motivation studies are plotted in Fig. 6.

Also, it was necessary to determine if the differences obtained are statistically significant. The normality test indicated that data from the survey were normally distributed. Therefore, the paired $t$-test with a 5% level of significance was calculated ($t = −1.761$ for attention; $t = −6.120$ for relevance, $t = −2.281$ for confidence, $t = −2.877$ for satisfaction, and $t = −3.613$ for ARCS). $P$-values less than or equal to 0.05 are considered significant and values greater than 0.05 as non-significant. Considering the null hypothesis, "there is no significant difference between pre-test and post-test scores":

- **$H_1$**: is *rejected*. We obtained $p = 0.081$; therefore, the difference of 0.19 is not significant regarding attention (A).
- **$H_2$**: is *accepted*. There is statistical evidence ($p<0.001$) to support that with SICMAR, a significant difference of 0.51 on students' relevance (R) is obtained.
- **$H_3$**: is *accepted*. The difference of 0.21 (4.08−3.87) is significant regarding the confidence (C) dimension with $p = 0.025$.
- **$H_4$**: is *accepted*. We obtained $p = 0.005$; hence, the difference of 0.3 is significant regarding students' satisfaction (S).

The magnitude and significance of causal connections between variables can be estimated using path analysis. We perform a path analysis to compute total effects among ARCS four dimensions and determine students' motivation. The diagram in Fig. 7 is the visual representation of the relationships between variables. The path coefficients (β) estimate the variance of the indicator that is accounted by the latent construct. The higher the value of β, the stronger the effect. The values calculated for the pre-test are shown above the arrows and the post-test values below the arrows. We also calculate the determination coefficients

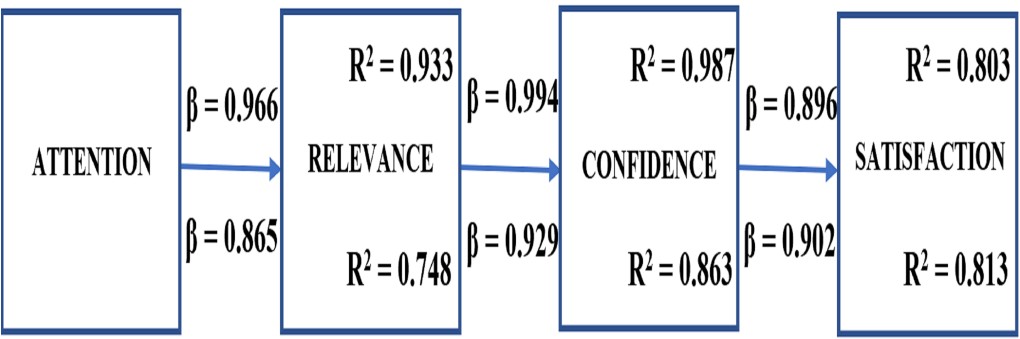

**Figure 7** Standardized path coefficients of the ARCS models.

**Table 5** Results for the practice tests.

| Test item | Pre-test | | Post-test | |
|---|---|---|---|---|
| | Correct | Incorrect | Correct | Incorrect |
| Simple interest ($I$) | 51 | 52 | 63 | 40 |
| Simple interest ($I$) | 69 | 34 | 66 | 37 |
| Principal ($P$) | 29 | 74 | 71 | 32 |
| Interest rate ($r$) | 24 | 79 | 68 | 35 |
| Time ($t$) | 28 | 75 | 75 | 28 |

($R^2$) to measure how close the data are to the fitted regression line (values must be greater than 0.2). The pre-test values are shown in the upper right corner and the lower right corner for the post-test.

It is noted from Fig. 7 that a significant direct effect exists from A->R, from R->C, and C->S with a significance level of 5% for both tests. Hence, the hypothesis:

- **H$_5$**: is *accepted*. Students increased their motivation using SICMAR. The value $M = 4.17$ obtained with SICMAR is greater than $M = 3.87$ obtained with the professor's lesson. The difference of 0.3 is statistically significant ($p<0.001$), representing a motivation increase of 7.75%. In summary, the mean values and the path analysis corroborate the motivation increase.

### Assessment of students achievement in practice tests

The professor reviewed the students' responses to emit the grade. An answer is correct only if the result and the procedure to obtain the response are good. Many students presented good results but a wrong procedure; these cases were qualified as incorrect. Since the test includes five items, each correct answer sums 20 points. Therefore, the final grade ranged from 0 to 100. A summary of the correct and incorrect responses for each test is shown in Table 5.

We obtained ($M = 39.02$ and $SD = 28.88$) for the pre-test and ($M = 66.60$ and $SD = 29.02$) for the post-test. Hence, an increase of 70.68% was observed on post-test grades when comparing with the pre-test. For the post-test, 25 students obtained the maximum

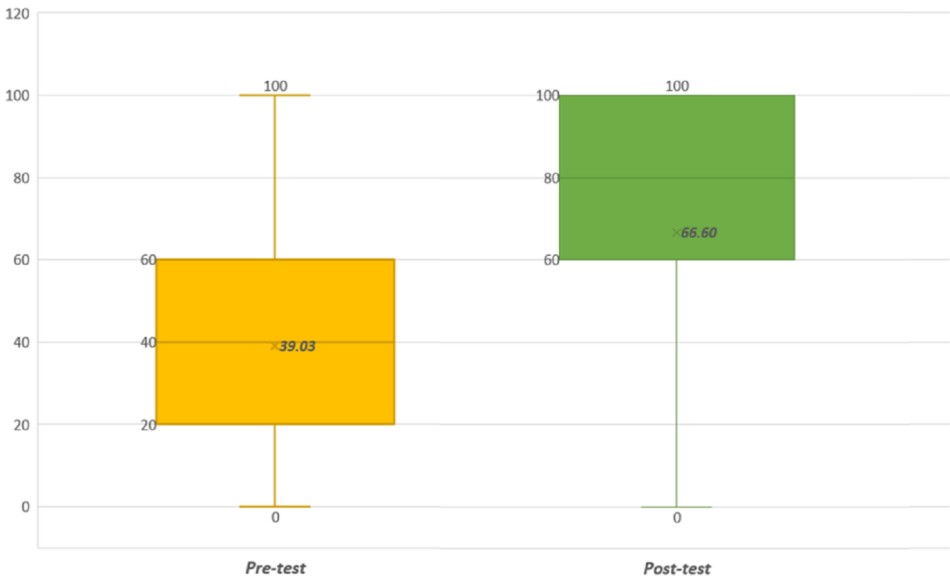

**Figure 8** **Results for the practice tests (pre-test and post-test).**

grade (100), and only four in the pre-test. Thirteen students (13.59%) obtained better grades on the pre-test than post-test. Moreover, 72 students obtained better scores for the post-test than the pre-test, while 18 students obtained the same score for both tests. In both sessions, women obtained better grades, with pre-test values ($M = 44.06$, $SD = 27.41$) and post-test values ($M = 76.61$, $SD = 20.41$). The pre-test values obtained for men were ($M = 32.27$, $SD = 30.56$), and for the post-test ($M = 53.18$, $SD = 31.38$). The plot (box and whiskers) of the scores obtained by students is illustrated in Fig. 8.

The test of Kolmogorov–Smirnov was employed to select the statistical analysis tool accordingly to the data distribution. The results obtained with a 5% level of significance using SPSS for the pre-test were: $Z = 0.162$, $p<0.001$, skewness $= 0.285$, skewness standard error $= 0.238$, kurtosis $= -0.885$, and kurtosis standard error $= 0.472$. For the post-test the results were: $Z = 0.192$, $p < 0.001$, skewness $= -0.675$, skewness standard error $= 0.238$, kurtosis $= -0.396$, and kurtosis standard error $= 0.472$. For both tests (pre-test and post-test) the results were: $Z = 0.109$, $p = 0.004$, skewness $= -0.329$, skewness standard error $= 0.238$, kurtosis $= 0.242$, and kurtosis standard error $= 0.472$, meaning that normality is not satisfied. Thus, the two paired Wilcoxon signed-rank test was utilized to observe if grade difference is significant ($Z = -6.129$, $p<0.001$, and medium effect size $d = -0.427$). Therefore:

- **H$_6$**: is *accepted*. Students obtained an average grade of 66.6 with SICMAR and 39.02 with the professor's material. The difference of 27.58 is statistically significant ($p < 0.001$).

### SICMAR technology acceptance assessment

We use AMOS software to examine the effects between observed and latent variables and the validity of the proposed hypotheses. The variables and the relationships between them were established considering *Miranda et al. (2016)* and *Hamidi & Chavoshi (2018)*.

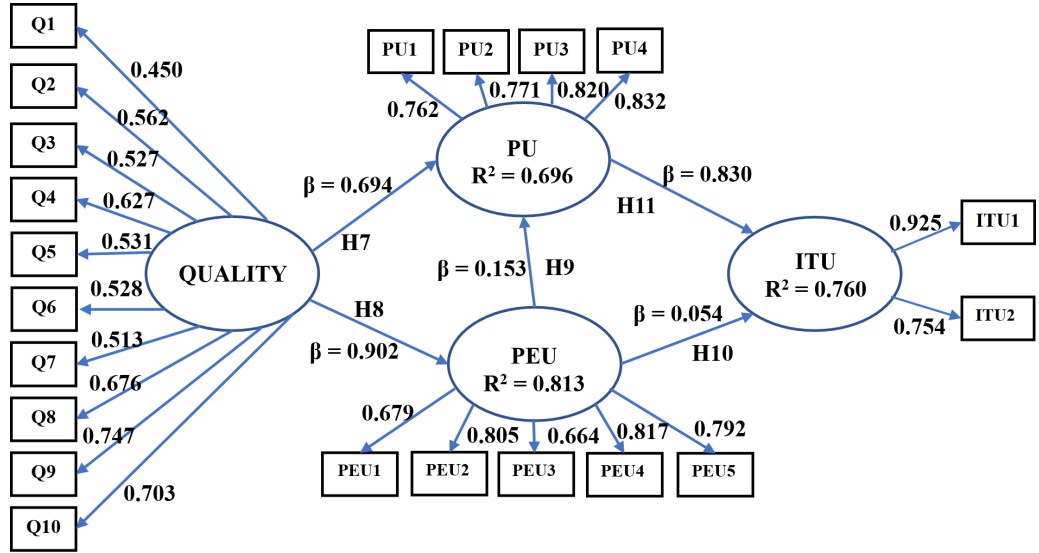

**Figure 9** The structural equation model and its standardized factor loadings.

The model in Fig. 9 comprises four latent variables (spheres) and 21 observed variables (squares). The relationships are symbolized with unidirectional arrows. The latent variable of quality is independent because no arrow is connected to it, and the remainder are dependent (at least one arrow was connected).

In structural equation modeling, only the identified (over, just, or under-identified) models can be estimated. Identification is the act of formally stating a model. We conduct the identification by computing the degrees of freedom ($DoF = 184$). When the $DoF$ is greater than 0, the model has more information than parameters to estimate. Therefore, our model is over-identified. Afterward, we calculate the sample variances and covariances to obtain the values that provide a reproduced matrix that best fit the observed matrix. A model fits the data well if differences between observed and predicted values are small. For this purpose, we employ the maximum likelihood method.

A summary of the values obtained is shown in Table 6. We expected that $\chi^2$/DoF ranged from 2 to 3, a GFI value near 1, and RMR closer to 0. Our model fulfills the conditions; therefore, we have a good-fitting model. Next, we compute the coefficients of determination ($R^2$) to measure the percentage of variance explained by the independent variables. The results are shown in Fig. 9. Values higher than 0.5 are considered good.

Then, we compute the standardized factor loadings and $p$-values for the observed variables. A summary of the results is shown in the third and fourth columns in Table 3. All the relations between observed variables to latent variables are accepted with a confidence of 1%. For the case of quality, the variables Q9 and Q10 related to markers were the most important. For the perceived ease of use, PEU2 and PEU4, which address the familiarity with the technology and manipulation of the prototype controls, obtained the greater values. Regarding the perceived usefulness, PU3 and PU4 were the most important, which

**Table 6  Structural equation model fit statistics.**

| Fit indices | Value obtained |
|---|---|
| DoF | 184 |
| $P$ | 0.000 |
| $\chi^2$ | 386.726 |
| $\chi^2$/DoF | 2.101 |
| Goodness of fit index (GFI) | 0.710 |
| Adjusted goodness of fit index (AGFI) | 0.635 |
| Standardized Root Mean Residual (RMR) | 0.080 |
| Comparative fit index (CFI) | 0.832 |
| Normed fit index (NFI) | 0.727 |
| Incremental fit index (IFI) | 0.836 |
| Parsimony goodness of fit index (PGFI) | 0.565 |
| Root mean square error of approximation (RMSEA) | 0.104 |

**Table 7  Path coefficients, direct, indirect, and total effects between the latent variables.**

| Paths | β | t | Standard error | p-value | Hypotheses Interpretation | Direct | Indirect | Total |
|---|---|---|---|---|---|---|---|---|
| Quality->PU | 0.694 | 2.487 | 0.247 | 0.013 | H7 Accepted | 0.694 | 0.138 | 0.832 |
| Quality->PEU | 0.902 | 6.819 | 0.121 | <0.001 | H8 Accepted | 0.902 | 0 | 0.902 |
| PEU->PU | 0.153 | 0.580 | 0.254 | 0.562 | H9 Rejected | 0.153 | 0 | 0.153 |
| PEU->ITU | 0.054 | 0.387 | 0.181 | 0.699 | H10 Rejected | 0.054 | 0.127 | 0.181 |
| PU->ITU | 0.830 | 5.301 | 0.211 | <0.001 | H11 Accepted | 0.830 | 0 | 0.830 |
| Quality->ITU | | | | | | 0 | 0.738 | 0.738 |

refers to the usability of SICMAR to learn and remember concepts. The highest value was obtained for ITU1, where students expressed their interest in keeping using SICMAR.

Finally, we compute the path coefficients (β), the *p*-values, and the direct, indirect, and total effects between variables (see Table 7). A direct effect is a relationship that exists between one variable to another. An indirect effect is a relationship between two variables mediated by at least one or more different variables. The sum of direct and indirect effects determines the total effect. Each direct effect is represented with a β in Fig. 9 and helps validate the hypotheses.

- **H₇**: is *accepted*. The quality effect on the perceived usefulness has $\beta = 0.694$ and $p < 0.05$. When the quality increases its standard deviation by one unit, the perceived usefulness goes up by 0.694 units, establishing a significant relationship with a confidence of 95%. Also, the quality establishes an indirect effect on the intention to use when it passes by the perceived usefulness.
- **H₈**: is *accepted*. The quality effect on the perceived ease of use has $\beta = 0.902$ and $p < 0.001$. When the quality increases its standard deviation by one unit, the perceived ease of use goes up by 0.902 units, establishing a significant relationship with a confidence of 95%.

- **H$_9$**: is *rejected*. The perceived ease of use effect on the perceived usefulness has $\beta = 0.153$ and $p = 0.562$. Therefore, the direct effect is not significant, with a confidence of 95%.
- **H$_{10}$**: is *rejected*. The perceived ease of use effect on the intention to use has $\beta = 0.054$ and $p < 0.699$. Therefore, the direct effect is not significant, with a confidence of 95%.
- **H$_{11}$**: is *accepted*. The perceived usefulness effect on the intention to use has $\beta = 0.830$ and $p < 0.001$. When the perceived usefulness increases its standard deviation by one unit, the intention to use goes up by 0.830 units, establishing a significant relationship with a confidence of 95%.

According to the results, the intention to use SICMAR is significantly affected by the quality and perceived usefulness. Students expressed their intention to use SICMAR due to the total effect of 0.738 encountered in the path Quality->PU->ITU.

***SICMAR quality assessment***

We conduct a study to determine if students considered SICMAR a good quality prototype. The scores obtained ($M = 3.93$ and $SD = 0.62$) demonstrate that students consider SICMAR a good quality prototype, as shown in Table 3. The minimum value obtained ($M = 3.16$) was regarding item Q5. Students considered that buttons are small, so we need to enlarge the buttons for better manipulation. The next minimum corresponds to Q10 ($M = 3.56$); hence, students could not easily manipulate the device and the markers simultaneously. The better results correspond to Q1 ($M = 4.45$) and Q6 ($M = 4.40$), which suggest that all the simple interest terms were included, and the velocity of response for computations was fast.

Data obtained from quality follow a normal distribution. Therefore, a one-sample $t$-test with a significance of 5% and a reference value of 3.8 was performed ($t = 2.126$, $p = 0.036$, and $d = 0.20$).:

- **H$_{12}$**: is *accepted*. A significant difference is obtained when comparing $M = 3.93$ with the reference value (3.8). Also, as mentioned in the TAM study, quality influences the students' intention to use SICMAR.

# DISCUSSION

From the results obtained, we observed that mobile augmented reality could be applied to financial mathematics, obtaining the benefit of increasing perception and user interaction with the environment; a non-AR application cannot offer those features.

The findings of our motivation study are consistent with those reported in the papers of Table 1. Mobile augmented reality changes the way in which students interact with the world. Moreover, the use of MAR increases students' motivation when learning about simple interest topics. Students expressed that using the fingers to insert the values for calculation and the interaction of the markers to define the inputs and outputs is attractive. According to the professor, students became more engaged during the post-test session. This is due to the different and interactive ways of presenting the information. Students expressed that MAR could turn a classic learning process into an engaging experience. Based on Table 2, the elements to increase students' motivation were: (i) regarding relevance, the

content, and style of the SICMAR explanations; (ii) regarding confidence, the organization of the information; (iii) regarding satisfaction, the design of the prototype (the interactive representations of time conversions, the 2D virtual objects, and how markers interaction determined the calculation to be computed).

Besides, students did not consider the quality of the contents, the organization of the information, and the variety of 2D models and interactions of SICMAR to keep their attention. The fact of using ICTs also influence the results obtained. Moreover, the younger participants felt more motivated with SICMAR, as expected.

According to *Loorbach et al. (2015)*, confidence influences students' persistence and accomplishment. Hence, it is crucial for motivation. In our post-test study, confidence (β = *0.902)* positively affects students' motivation. The main differences of our findings with the works by *Estapa & Nadolny (2015)*, *Cascales et al. (2017)*, and *Ibáñez et al. (2020)* were that our sample size is the biggest, we used RIMMS instead of IMMS (since there are few items, students are less worn), and that we utilized path analysis. In summary, the statistical results indicate that students who used SICMAR significantly increase their motivation scores (7.75%) compared with scores obtained in the professor's lesson.

By observing Fig. 8, students performed better when answering the practice exercises using SICMAR compared with the professor's lesson's answers. An increase of 70.76% was observed. Regarding simple interest computation, students using SICMAR performed better for the first question but not for the second (see Table 5). All the students with incorrect answers to these questions for the pre-test failed to convert the time unit. On the other hand, only ten students in the post-test failed to convert the time unit. Mistakes such as not including/describing the procedure to solve the problem, not copying the correct answer, and wrong selection of markers were the most common.

Regarding exam questions 3–5 in Table 5, it is notable that the performance increases in students using SICMAR. In these questions, the terms to compute the solution must be cleared. All the students with incorrect answers for the pre-test failed to convert the time unit. On the other hand, half of the incorrect responses for the pre-test were due to conversions. The remaining mistakes were due to not including the procedure to solve the problem or not copying the correct answer.

The works by *Estapa & Nadolny (2015)*, *Tobar, Fabregat & Baldiris (2015)*, and *Ibáñez et al. (2020)* reported about students' achievements; however, they measured the time used to execute the tasks, unlike our proposal that quantified the answers of practice exercises. *Purnama, Andrew & Galinium (2014)* reported an increase of 17% in the learning process; unfortunately, the way it was measured was never explained. *Coimbra, Cardoso & Mateus (2015)* presented only qualitative preliminary explanations about math learning enhancing. Therefore, we cannot provide comparisons against literature works.

None of the works in Table 1 utilized the TAM; hence, we cannot offer comparisons. However, the path Quality->PU->ITU determined the students' intention to use SICMAR. Students gave the highest quality scores to the concepts explained, the calculation speed, the results, the colors, and the legibility of texts displayed. On the other hand, the question regarding the size of the buttons obtained the lowest score. We observed that smartphone users offered all the comments about the small size of the buttons. This can lead to conduct

future research to obtain insights into how a mobile device's screen size influences the perception of the AR experience. Students considered SICMAR useful for learning, and it helped to remember the concepts related to simple interest. Finally, students expressed SICMAR quality enough to use the prototype continuously.

## Lessons learned

Our augmented reality educational prototype serves as an alternative tool to learn the simple interest topic, but it cannot replace the teacher. Professors will continue looking for tools to improve the teaching-learning process. However, many times, teachers are not willing to make the efforts to create the tools, and frequently they do not have the computer skills to develop them because a usable app is challenging to create. The software to rapidly create augmented reality experiences does not offer all the resources needed to explain complex science topics.

Augmented reality causes enjoyment in students and a desire to repeat the experience. Although not complex 3D models were needed to represent the augmented reality for SICMAR, this alternative representation of the real phenomena causes motivation to the students. Even when the literature has been asserted that augmented reality can be exploited in any field, we recommend choosing application areas where it is needed that 3D models show different views of the objects. Even though 3D models are the base of augmented reality, it is still difficult to explain how the computer-based models inserted into the real scene increase student achievement.

Some students expressed that prolonged use of SICMAR slows down and warm up the device. We know that this problem is common when a mobile device is used for a considerable time. However, we are not sure if this problem is accentuated due to the execution of our prototype's complex routines. This leads to inviting developers to conduct a thorough review to detect routines that can optimize the processor usage, look for native development, or test another framework for coding.

Remarkably, we detected that students commit mistakes in handling the five markers to manipulate SICMAR. For example, they selected the principal marker when the correct one was the amount. Therefore, SICMAR cannot help students understand the problem stated and neither identify the concepts involved. The solution to the issue of problem understanding is still an open challenge. We recommend to developers avoid combinations of many markers to trigger augmented reality.

If professors and students disagreed with testing SICMAR, they would still be thinking that paper, blackboard, books, slides, or computer-based content are the unique resources to learn. With the findings obtained, we observed the potential of augmented reality for educational settings. The school administrators must be convinced that augmented reality is an educational tool that they should provide for the students. Moreover, schools must invest effort into implementing more resources based on augmented reality to the complete curriculum. With this, it will be possible: (i) to observe the real impact of augmented reality on students; (ii) have tracking about the usability of the resources; (iii) detect the moment when the interest is lost; (iv) establish if the impact was due only to the novelty; and (v) to know what happens with concentration, and cognitive load of the students.

## Limitations

After experimentation, we note some limitations in our study. Because we conducted a quasi-experimental study, our data might be biased. Therefore, the reported findings should be replicated with an experimental study (using control and experimental groups). We cannot know if students change their behavior because they are aware of participating in a research study. Only one financial mathematics professor used SICMAR, so the positive comments regarding usability may change as more teachers are involved. Some students focused their attention on the application and not on the essential parts of the topic to learn. This fact is known as the attention tunneling effect, which can explain why some students scored lower using SICMAR. Also, not all students felt comfortable using SICMAR, which offered clues that for some person it could be challenging to use ICTs. Moreover, the issues related to gender were not analyzed in depth, which is currently a trend in the AR field.

## Future research

Extensions of the proposed study include the improvement of the interaction environment, a larger sample of students, the measurement of cognitive load, to involve more financial mathematics teachers, and the implementation of other topics about financial mathematics. It is desirable to study in-depth the case of students which grade has decreased using SICMAR. The worst case is a student who obtained a grade of 80 with professor lesson and zero with SICMAR. Also, in-depth analyses of the individual answers to the tests and the questionnaires will be conducted. A critical review of the interface design should be performed. Further research is necessary to determine the content that must be added to SICMAR to keep students' attention. Finally, it would be recommendable to run a pilot study with Microsoft Hololenses to observe if the possibility of not clicking on screens increases students' motivation and achievement.

## CONCLUSIONS

In this paper, the SICMAR prototype based on augmented reality was introduced to verify its effects in learning the concept of simple interest to undergraduate students of financial mathematics. To the best of our knowledge, concepts such as principal, amount, time, interest rate, and simple interest are considered fundamental to promote students' financial education. SICMAR was tested in a real university setting to assess its quality, students' motivation using ARCS, the achievement by answering practice exercises, and technology acceptance with extended TAM. The results obtained from tests with 103 participants revealed that the undergraduate students were interested in using SICMAR frequently because of its quality, they were motivated to learn the simple interest topics and increased their achievement in answering practice exercises. All this leads to concluding that SICMAR is a valuable complementary tool to learn the concepts related to simple interest computation.

### Funding

The Dirección General de Educación Superior de la Secretaría de Educación Pública de México granted their support to the project "Introducción de la Realidad Aumentada en las escuelas de negocios" developed by the group "Tecnologías de Información y Comunicación en las Organizaciones (UANL-CA-368)". The funders had no role in study design, data collection and analysis, decision to publish, or preparation of the manuscript.

### Grant Disclosures

The following grant information was disclosed by the authors:
Dirección General de Educación Superior de la Secretaría de Educación Pública de México: UANL-CA-368.

### Competing Interests

The authors declare there are no competing interests.

### Author Contributions

- Laura Alicia Hernández Moreno conceived and designed the experiments, performed the experiments, analyzed the data, performed the computation work, authored or reviewed drafts of the paper, and approved the final draft.
- Juan Gabriel López Solórzano conceived and designed the experiments, performed the experiments, analyzed the data, performed the computation work, prepared figures and/or tables, and approved the final draft.
- María Teresa Tovar Morales conceived and designed the experiments, performed the experiments, prepared figures and/or tables, and approved the final draft.
- Osslan Osiris Vergara Villegas conceived and designed the experiments, analyzed the data, prepared figures and/or tables, authored or reviewed drafts of the paper, and approved the final draft.
- Vianey Guadalupe Cruz Sánchez conceived and designed the experiments, analyzed the data, authored or reviewed drafts of the paper, and approved the final draft.

### Ethics

The following information was supplied relating to ethical approvals (i.e., approving body and any reference numbers):

Universidad Autonoma de Ciudad Juárez, particularly by the Institute of Engineering and Technology emmited approval to carry out the study.

### Data Availability

The responses to our surveys are available in the Supplemental File.

### Supplemental Information

Supplemental information for this article can be found online at http://dx.doi.org/10.7717/peerj-cs.618#supplemental-information.

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
