# Peer review of "Effects of using mobile augmented reality for simple interest computation in a financial mathematics course"

_PeerJ Computer Science, doi:10.7717/peerj-cs.618_

## Round 0.1 · original submission · Major Revisions

· Academic Editor

Major Revisions

As you will see from the comments by the reviewers your submission is far from acceptable for publication in this journal. I am grateful to the reviewers for their detailed suggestions for improvements.
I recommend that you take longer than the suggested 30 days to improve the paper substantially. My decision of "Major Revisions" means that if you are not able to improve the paper sufficiently before resubmission then it is likely to be rejected. Given the relevance of the work, I hope that you will be able to provide all the proposed improvements.

·

Basic reporting

Good language, easy to read, structure is fine but for a suggestion of revision in the details below.

Experimental design

Fine but but a few suggestions below.

Validity of the findings

Fine.

Additional comments

Instead of writing a lengthy text In prose, I provide a detailed list of comments and recommendations:
- The topic of your paper is extremely specific, yet it is quite interesting.
- You claim that students "show no interest in financial education" but looked at "undergraduates enrolled in financial mathematics courses". Why should anyone without an interest in finance take a course in financial mathematics? As part of not directly related degree programs? Please explicate.
- The hypotheses are fine. I do not think they should be placed in the introduction, though, where I would only expect the research question. I would develop the hypotheses in a section on research design.
- Nice to explicate your contributions.
- Your literature study is too short, particularly concerning general work on the use of VR / AE in education. For the VR side, I recommend the following article as a starting point:
Radianti, J., Majchrzak, T. A., Fromm, J., and Wohlgenannt, I. (2020): A systematic review of immersive virtual reality applications for higher education: Design elements, lessons learned, and research agenda, Computers & Education, 147, 103778, Elsevier
I think you did a great job regarding related studies, but you could put more detail into drawing the "broader picture" of related research activities.
- I am tempted to suggest renaming "Materials & Methods" to "Research Design", to include hypothesis building (possibly also adding justification for the different hypotheses), and to make it more a classical section on how the research is conceptualized and carried out.
- I think it would be nice to get more details on the development. You may consider adding a model of the tool.
- The survey detail should be described in more detail.
- Selection of the participants should be described in more detail.
- The presentation of the results is fine.
- The discussion should be much extended. I suggest to at least have subsections that
-- discuss the lessons learned from your study and how this related to the literature and your expectations,
-- give implications for research and state possible contributions to theory, thereby generalizing your findings beyond its narrow domain,
-- scrutinize implications for practice, such as advice for educators, and
-- name limitations of your work.
- You may also give further research directions.
- The reference list is surprisingly short for the topic at hand. I think you should check if you can relate to works that use AR for education, even if domains and settings are different. Quite possibly this can yield valuable insights.
- Figures that are vectors by source must be inserted in vectorized form. This applies to Figure 1, Figure 8, and Figure 9.
- The tables are very nice and helpful.
- The supplementary materials make a very good impression.

·

Basic reporting

The English language is not clear enough. It contains grammatical or syntactic errors (highlighted in purple in attachment) or vague phrasings that are not sufficiently informative (highlighted in blue in attachment).

The set of references is extensive, however, the authors often do not clearly indicate which information or decision come from the literature (with or without adaptations) or are assumptions. For instance, the model in Figure 9 come from Hamidi & Chevoshi, with modification (omissions). References to the literature are sometimes unspecified (e.g., "recent studies" mentioned p.10 without citation). For instance, reference to Davis (1989) is missing:
Davis, F. D. (1989). Perceived usefulness, perceived ease of use, and user acceptance of information technology. MIS quarterly, 319-340.

Acronyms are used too often, and sometimes unnecessarily, without recapping their meaning. It makes it difficult to read the paper. The acronym for Perceived Ease of Use is inconsistent (PEOU Vs PEU). Finally, null p-values are reported (p = 0.000) which is misleading and inexact (better say p < 0.001).

Equations (1-6) are highly redundant, and may not reflect the equations used in practice, from a human perspective. Mentioning only equations (2) and (5-6) should be sufficient. The meaning of the equation terms should be stated in advanced, e.g., as an introduction to the topic at hand. In particular, the term "simple interest" is barely explained or understandable until the equations are shown.

Hypotheses are not sufficiently explained before being stated. For instance, it is unclear what "positive effects" mean until SEM is mentioned. As well, Hypotheses 12 mentions a threshold of 3.8 without any justification. It seems arbitrary until it is explained later.

Experimental design

The experimental design relies on solid statistics, however it does not provide specific insights into what makes the interface successful or not. There is no comment on the interface design itself. Furthermore, the UI design could be applied on simple computer screen, and Augmented Reality (AR) technology is not providing extra features. It may have a positive effect due to its novelty, but the interface may be more usable on simple computer screen.
This research omit such consideration, hence it is not possible to claim that the positive results are due to using AR or using the SICMAR design. Similar positive impacts may be observed using simple computer screens, compared to learning with a teacher.

Finally the experimental design has a crucial flaw. Students are first exposed to pedagogical content with a teacher, before they are again expose to contents on the same topic using SICMAR. No control group is set apart, and exposed again to pedagogical content without using SICMAR. Hence the observed positive impacts may be due to the students' learning curve, not to the use of SICMAR. Similar positive impacts may be observed even if students had practiced again with a teacher.

Validity of the findings

The authors mention exaggerated claims and interpretations of their findings. It is most important to state the limitations of the study. Besides limitations due to the experimental design, the data analysis has additional limitations.

Other limitations:
- The text of the questionnaire contain English mistakes and is sometimes vague or difficult to read. The poor quality of the English language may have affected the study. (e.g., misunderstanding, fatigue).
- The statistical analysis is extensive, and is sufficiently explained, but it remains quite difficult to read due to the phrasing and grammar.
- The individual data points could be plotted relatively easily, and would make it easier to review the findings.
- Answers to single questions (e.g., A1, A2, etc...) should be compared individually, i.e., answers with or without SICMAR.
- The formula to calculate the "Totals" in Tables 2, 3, 5 is not specified. Furthermore, it is incorrect to call these totals, as they are means (since their range remains within 1-5).

When analysis the grades of the exam:
- The text of the exam is not provided.
- The answers to individual questions are not analysed (e.g., to identify what is particularly difficult to learn)
- The progress of individual students is not analysed, just averages are provided. However, the author mention that some student grades have decreased with SICMAR (how many? by how much?).

Additional comments

This research investigates a promising technology that may enhance the learning process of students. However, the research design has crucial methodological limitations, and the findings are not specific enough to provide guidance or insights for future work (i.e., to inform the design such tools, rather than observing general improvements of student's motivations or exam results). To be more valuable, the research should be continued, e.g., with in-depth analyses of the answers to the exam and to the questionnaires, and by making a critical review of the interface design (i.e., in relation to the analysis of the exam and questionnaire).

The annotated document attached uses the following color code:
- Yellow: my own notes, please ignore these
- Blue: vague phrasing, to hard to understand or not informative enough
- Purple: English mistakes
- Pink: statements that are arguable from a scientific perspective, or that are not justified enough to be convincing.
- Green: Good points, but not all good points are highlighted.

---

## Round 0.2 · Minor Revisions

· Academic Editor

Minor Revisions

Firstly, my apologies for the delay in getting back to you.
The submission has been significantly improved on the basis of the reviewers feedback - thank you for this. One reviewer, however, still has points that should be addressed before going to publication.

·

Basic reporting

Good language, easy to read, structure is fine.

Experimental design

Fine.

Validity of the findings

Fine.

Additional comments

Dear authors,

thank you for the revision and for the good overview of changes in the rebuttal document. I think you did an excellent job in commenting on and addressing the reviews.

The amount of changes is impressive. You obviously took the revision very serious, which is appreciated. Although it is somewhat uncommong to accept R1 after a major revision "as is", I recommend doing so. The only strong suggestion that I have is inserted Figures 6 to 9 in vectorized form (I am not sure about Figure 3, it might be possible as well).

·

Basic reporting

The English has (almost) no mistake, the references are complete, and the methods are described in detail. The sections on "Lessons learned" and "Limitations", and the additional literature review (incl. Table 1) are very good additions.

However, there are still sentences that are not precise enough (details below). It makes the discussion state claims with too large a scope, and that are not fully supported by the study in the paper. Namely:

- L.674-675, "Students use their fingers to insert the values and uses the markers to define the inputs and outputs. As a result, students' motivation to learn simple interest increases.": The paper provides no evidence that a specific way to use fingers is what increased motivation.

- L.678, "students perceived learning as a game": The paper does not provide evidence to this, i.e., no questions asked to participants targeted this aspect in particular. Perhaps oral was collected, but this is not reported.

- L.683-684, "students did not consider the contents of SICMAR as sufficient to keep all the attention": The absence of significant impact on attention, as measured with the questionnaire, cannot be directly linked to the sufficiency (sic) of content.

- L.715-716, "Students considered the [...list of features...] as the critical features to determine quality": The word "consider" is ambiguous. To be precise, students gave the highest quality scores to these features. It does not mean that they identified these as "critical features to determine quality". The authors of the paper are the ones determining quality using these features (and others in the questionnaire), perhaps students would consider other features as critical for quality.


Line by line details:
- L.65-66, "principal sum/balance": the terms can be unclear for a reader with no expertise in finance.
- L.154-155, "Qualitative research is the most common theory-base employed, followed by the nonparametric Wilcoxon signed-rank test": Comparing qualitative research with a statistical test is odd. What matters is what is tested, more than the type of test.
- L.192: the definition of "interest rate" is imprecise. In particular these words in [...] : "the amount charged [on top of] the principal [for the use of assets] (expressed as a [percentage])". The definition does not mention of the time dimension, and it's a percentage of what? Also equation (4) is not a percentage, but that's a detail. So it's rather, e.g., "a fraction/percentage of the Principal (charged) per unit of time".
- L.193: the definition of "time" is also imprecise, e.g., "the period of the [financial operation]" could be phrased as "the time period over which the interest rates apply/are charged".
- L.203, "Eqs. (1) to (6) are expressed in years": the interest rate is not expressed in years, i.e., it is not the unit. To be precise, one should rather say "use year as the time unit".
- L.297-298: "much quality" -> "good/sufficient quality"
- L. 333 & 670, "improves the user perception and interaction with the real world": This is rather vague, what is improved exactly?
- L. 336: "gadgets" -> "devices"
- L.362, "The context and world model subsystem includes...": I'm not sure why the terms "context" and "world model" are used, nor why the element listed thereafter fit together.
- L.420: "two five items tests" -> "two five-item tests". 
- L.432, "translated": You should mention the language was Spanish, and so for the whole experiment.
- L.583, 584, 586: "were (..." -> "were: ..." and L.586 is missing "[the results] were".
- L. 596, "pre-test-post-test: pre-test and post-test.
- L.592, "3.33 times": it is vague, and the unit is not that of the grades (total of 100 points). The reader can only understand this number by looking at Fig. an dividing the mean by 20 (20 points per question). You should mention the "average grade of 66.6".
- L.602: "state" -> "stating"
- L.673, "the way on how" -> "the way in which"
- L.677 & 718: "determined" -> "expressed"?
- L.682: "bi-directional models" -> "2D virtual objects"
- L.698, 699, 704: "failed in/due the period conversion" -> "failed to convert the time unit" (3 occurrences)
- L.699-700 + 705-706: "not include" -> "not including/describing" and "not copy" -> "not copying"
- L.702, "questions 3-5" add "in Table 5" and/or "exam question". + "notable [that] the performance increase[s]".
- L.709: "qualified" -> "quantified"
- L.768: "some persons could be challenging using ICTs" -> "for some person it could be challenging to use ICTs" and "in-depth-analyzed" -> "analyzed in depth"
- L.775: "students that decrease the grade" -> "students which grade has decreased"
- L.776: "cero" -> "zero"
- L.781: "not using the hands" -> "not clicking on screens" (hand gestures are still needed. 
- L.793: "conveys to conclude" -> "leads to concluding"

Experimental design

No further comment.

However, another detail concerns the type of mobile devices used with the AR app. Figure 5 shows a tablet, and I was under the impression that mobile phones were used. The type of device (e.g., screen size) can influence the results of the experiment. This should be clarified: e.g., did all students used a tablet? a phone? or a mix of both?

Validity of the findings

The section added on "Limitations" and the author's rebuttal seem to address most of my concerns. However, my concern with exaggerated claims holds, but I'm willing to consider that this is due to issues with how claims are written down (see section 1 above).

Furthermore, regarding the 36 students that were removed from the study (fro not completing the questionnaires): this is a considerable number (25% of the participants). This could impact the validity of the study. For instance, these students may have had issues with prototype, that could explain their incomplete answers to the questionnaires. If so, their negative feedback would not be reported (but the authors could check if their partial answers indicate such effect).

---

## Round 0.3 · accepted · Accept

· Academic Editor

Accept

The paper still has minor issues with language. I recommend a final read-through, preferably by an English language expert, before going to press.

I noticed a duplicate paragraph -

69 In financial mathematics, interest is calculated as simple interest or compound interest; the
70 former determines how much interest to apply to a principal balance, whereas the latter is the
71 addition of interest to the principal sum of a loan or deposit (Hastings, 2015).